



# Development of an under-ice river discharge forecasting system in Delft-Flood Early Warning System (Delft-FEWS) for the Chaudière River based on a coupled hydrological-hydrodynamic modelling approach.

Kh Rahat Usman[1], Rodolfo Alvarado Montero[2], Tadros Ghobrial[1], François Anctil[1], and Arnejan van Loenen[3]

[1]Department of Civil and Water Engineering, Laval University, Québec City, Québec, Canada
[2]Department of Civil Engineering, Schulich School of Engineering, University of Calgary, Calgary, Alberta, Canada
[3]Deltares, United States and Canada

*Correspondence to*: Kh Rahat Usman (kh-rahat.usman.1@ulaval.ca)

**Abstract:** Year-round river discharge estimation and forecasting is a critical component of sustainable water resource management. However, in cold climate regions such as Canada, this basic task gets intricated due to the challenge of river ice conditions. River ice conditions are dynamic and can change quickly in a short period of time. This dynamic nature makes river ice conditions difficult to forecast. Moreover, the observation of under-ice river discharge also
remains a challenge since no reliable method for its estimation has been developed till date. It is therefore an active field of research and development. The integration of river ice hydraulic models in forecasting systems has remained relatively uncommon. The current study has two main objectives: first is to demonstrate the development and capabilities of a river ice forecasting system based on coupled hydrological and hydraulic modelling approach for the Chaudière River in Québec; and second is to assess its functionality over selected winter events. The forecasting
system is developed within a well-known operational forecasting platform: the Delft Flood Early Warning System (Delft-FEWS). The current configuration of the systems integrates (i) meteorological products such as the Regional Ensemble Prediction System (REPS); (ii) a hydrological module implemented through the HydrOlOgical Prediction LAboratory (HOOPLA), a multi-model based hydrological modelling framework; and (iii) hydraulic module implemented through a 1D steady and unsteady HEC-RAS river ice models. The system produces ensemble forecasts
for discharge and water level and provides flexibility to modify various dynamic parameters within the modelling chain such as discharge timeseries, ice thickness, ice roughness as well as carryout hindcasting experiments in a batch production way. Performance of the coupled modelling approach was assessed using "Perfect forecast" over winter events between 2020 and 2023 winter seasons. The root mean square error (RMSE) and percent bias (Pbias) metrics were calculated. The hydrologic module of the system showed significant deviations from the observations. These
deviations could be explained by the inherent uncertainty in the under-ice discharge estimates as well as uncertainty in the modelling chain. The hydraulic module of the system performed better and the Pbias was within ±10%.

## 1. Introduction

Water resource management agencies across Canada have developed forecasting systems for efficient management of water resources throughout the year; however, the winter season presents a special challenge to all agencies when the



natural streams are affected by an ice cover, rendering the under-ice discharge estimates subjective to expert judgment, interpolations and theoretical methods (Turcotte and Morse, 2017). These practices introduces significant uncertainty and inaccuracy in the winter records (Dahl et al., 2019). Moreover, most of these methods are passive in nature i.e., the winter flow records are not produced in real-time. Some agencies produce preliminary estimates and short-term discharge forecasts, but final validated values are produced and published at the end of the season.

River ice forecasting essential to flood management during winters for several basins across Canada (Pietroniro et al., 2021). In winters, flooding is usually associated with the formation or release of ice jams and can occur even at discharges much lower than those in open water conditions (Beltaos and Prowse, 2001; Beltaos, 2021). Apart from flood risk assessment, river ice forecasting also provides essential information for water resources management especially for the hydropower sector. River ice modelling is a critical component of any forecasting system developed

in cold climate regions, as it considers the influence of the ice cover on the flow dynamics and flooding events (Montero et al., 2023). River ice conditions are dynamic and can change rapidly over a short time period, hence, it remains an active area of research and development (Belvederesi et al., 2022).

Accurate estimation of under-ice discharge has remained a challenge for the river ice community and forecasting agencies. Therefore, various attempts have been made to come up with methods or techniques to improve practices

and procedures. In Canada, Water Survey of Canada (WSC) assumes major responsibility for this task. The WSC has established methodologies for varying conditions and sites. This comprises of collecting preliminary data through hydrometric stations, estimating instantaneous discharge from rating curve or index velocity method (Healy and Hicks, 2004). At some sites rating curves based on ice conditions have also been developed. The WSC also conducts direct measurements; however, they are low in frequency. At the end of the season a post processing of the entire winter

dataset is performed by WSC technicians using methods such as Recession constant, Graphic Interpolation, comparison of hydrograph, and backwater adjustment models. Other agencies across Canada have adopted similar practices comprising of handful of direct measurements and post processing instantaneous data (Turcotte and Morse, 2017).

Hydrological modelling approaches have been tested for the simulation of winter discharge (Hamilton et al., 2000;

Turcotte et al., 2005; Levesque et al., 2008). Hamilton et al. (2000) applied a conceptual hydrological model (a variant of the HBV model (Bergström, 1995)) for simulating daily discharge estimates in M'Clintock River watershed, Yukon, and concluded that hydrological modelling produced reasonable estimates for winter discharge but failed to capture discharge variability over the season. Turcotte et al. (2005) compared the performance of hydrological modelling to a data driven Neural network model. Their study found that both hydrological model and the Neural

Network model performed equally well before snowmelt started influencing the streamflow during the winters and concluded that the skill of hydrological modelling could be improved if winter streamflow gauging data is used for model calibration. A study by Levesque et al. (2008) also showed potential of hydrological modelling for winter discharge estimation where a process based Soil & Water Assessment Tool (SWAT) model was calibrated for streamflow estimation in two small catchments in Southern Québec. Hicks and Healy (2003) investigated the viability

of using hydraulic modelling (gradually varied flow conditions) to determine under-ice discharge, based on data obtained from the Mackenzie River and the Athabasca River. The study found that hydraulic modeling has the





potential to provide a substantial increase in accuracy compared to conventional approaches, with maximum error of less than 3% in discharge determined by this method. However, the authors cited a potential limitation to this approach was the estimation of the ice cover thickness. Ice cover thickness is not uniform along a river as well as across a cross

section. This may induce some degree of uncertainty in this method of estimating under-ice discharge.

The interaction of ice with river hydraulics is a complex process that depends on a number of factors such as surface ice concentration, the type of the cover, and the hydraulic regime of the river (Montero et al., 2023). Research into these processes have lead to the evolution of several river ice models (e.g., RIVICE, River1D) describing different river ice processes such as thermal exchange, ice formation processes, and ice jams hydraulics (Blackburn and She,

2019; Rokaya et al., 2022). Although these models enhance our comprehension of various river ice processes, their data-intensive nature for calibrating various parameters limits their applicability in operational forecasting systems. Therefore, operational forecasters prefer to adopt a simplistic approach rather than intense process-based modelling. HEC-RAS is a hydraulic modelling software that provides a suitable option since it can simulate simple ice covers and ice jams hydraulics. It is widely used by hydrotechnical engineers, and is available open source with a user friendly

interface and documentations (Beltaos and Tang, 2013). For simple ice cover hydraulic modelling, HEC-RAS has limited data requirements mainly ice thickness and under ice roughness at each cross-section which makes it applicable in an operational setting (Montero et al., 2023).

Recently coupled modelling approaches are gaining popularity among researchers in field of flood forecasting (Mai and De Smedt, 2017; Liu et al., 2019; Bessar, 2021; Rokaya et al., 2022). This approach consists of connecting

different types of models together to create an integrated modelling chain for forecasting floods. Usually this consist of a hydrological model serving as a streamflow prediction layer and subsequently informing a hydraulic model which then maps out flooding extent. Nevertheless, more layers of complexities can be added to the approach such as coupling meteorological prediction systems (MPS) to hydrological systems (Zhijia et al., 2004; Cattoën et al., 2016). Within these systems another degree of sophistication is added by adopting the ensemble approach that helps mitigate

uncertainties (Bessar et al., 2021) arising from different sources in the modelling chain and communicate the range of possible outcomes, providing decision-makers with a more comprehensive understanding of the associated risks.

Despite recognizing the impact of river ice on water resource management and ice jam related flooding, operational forecasting systems that integrate a river ice modelling component are rare (Montero et al., 2023). A recent example of such an integration is the lower Churchill River in Labrador, where river ice modelling is embedded into an

operational forecast. The system forecasts flows on the river using the hydrological model HEC-HMS and estimates water levels using the hydraulic models HEC-RAS (for open water) and RIVICE (for ice conditions) (Lindenschmidt et al., 2021). This system operates in deterministic mode without the application of data assimilation which makes it susceptible to forecasting errors and uncertainties. Moreover, it does not describe any framework to update hydraulic parameters such as the under-ice roughness which evolves over the season.

The current study investigates the potential of an ensemble based coupled hydrologic and hydraulic modelling approach to address the winter hydrometry challenge in an operational forecasting setting. Hence the study has two main objectives: *i)* to demonstrate the development and capabilities of an ensemble based coupled hydrologic-hydraulic modelling operational forecasting system, *ii)* to assess the functionality and performance of this system for



forecasting under-ice discharge and associated water levels using some selected events. The couple modelling system
was developed within the Delft Flood Early Warning System (Delft-FEWS) platform (Werner et al., 2013) for the
Chaudière River in Québec, Canada. The paper first describes the case study area and available data. It then provides
a description of components of the Delft-FEWS system and its capabilities to integrate different input data sources to
each model. Finally, the performance of the system is demonstrated based on several under-ice hydrologic events
observed in the winters between 2020 to 2023.

**2. Case study and data**


**2.1 Description of watershed**

The description of the watershed for this study has been presented in Montero et al. (2023). A short summary is
presented here. The Chaudière River basin is selected as the study basin. It is located South-East of Québec City,
Canada. The river has its source at Mégantic Lake and drains into the Saint Lawrence River near the Town of Lévis,
with a river reach of 188 km and a total watershed area of 6694 km². For this study, the Chaudière River basin has
been divided into three sub-catchments. Table 1 provides the main characteristics of the three sub-catchments. Figure
1 presents the study site.

**Table 1 Characteristics of the Chaudière Basin sub-catchments**

| Sub-Catchment | Area (km²) | River stationing | Hydrometric station | Mean river slope (m/km) |
|---|---|---|---|---|
| Upper Chaudière | 3085 | 188.0 – 106.78 | 023429 | 2.5 |
| Intermediate Chaudière | 5,820 | 106.78 – 25.60 | 023402 | 0.5 |
| Famine | 714 | 101.37 | 023422 | - |


The lower Chaudière (Chainage 25.60 to 0.0 km) is not included in this study since this section is steep and does not
pose any serious ice jam flooding hazards. Ice jam floods are common in the Intermediate Chaudière where the river
profile is flatter and exhibits several meanders, islands and bridges (Montero et al., 2023). The Comité de bassin de la
rivière Chaudière (COBARIC) operates four water level monitoring stations along the Intermediate Chaudière located
near major urban communities (labeled as COBARIC stations in Fig. 1). These locations are Beauceville (ch. 87.5
km), Saint Joseph (ch. 71.5 km), Valée Junction (ch. 61.8 km) and Sainte Marie (ch. 52.1 km) (Montero et al., 2023).
A detailed description of the Chaudière catchment and the river ice characteristics can be found in Ghobrial et al.
(2023).

The climatic regime dominant in Southern Québec is classified as humid continental (Dfb) according to the Köppen
classification (Kottek et al., 2006). The average monthly temperatures in the Chaudière River basin show a high degree



of seasonal variation with below freezing temperatures (average -6 °C and min -12.6 °C) between November-March and moderate temperatures (average 18 °C, high 25 °C) during the summers (Montero et al., 2023). The average annual precipitation is estimated to be 1031.5 mm shared between rain (824.9 mm) and snow (202.6 mm) (MELCCFP, 2023). Precipitation analysis of the catchment does not show significant variation between different months however,

between June and August the intensity of precipitation is slightly higher (Montero et al., 2023). The hydrological regime for the catchment can be classified as nivo-pluvial since the spring flood caused by snowmelt and precipitation is dominant, followed by flooding in autumn season due to excess rainfall (Ricard et al., 2023).

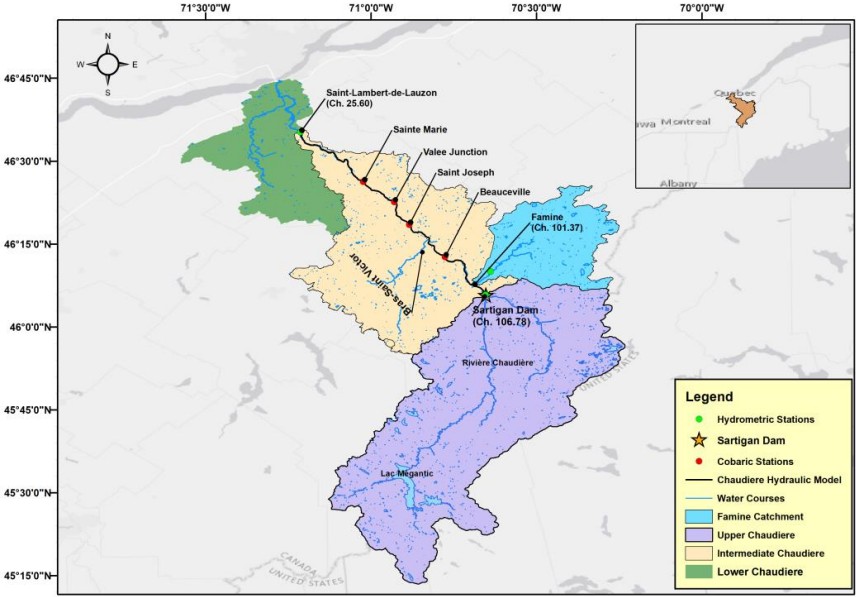

**Figure 1: Chaudière River basin and its sub-catchments based on hydrometric control points (green dots). The hydraulic model is developed for the Intermediate Chaudière between Sartigan Dam and Saint Lambert-de-Lauzon (black line). Figure adopted from Montero et al. (2023).**

**2.2 Data**

The data used in this study can be broadly classified into three main categories i.e. (i) Meteorological data, and (ii) Hydrometric data and (iii) Geographic data. These three different types of datasets were incorporated into the Delft-FEWS system. The meteorological dataset consists of observed and forecasted precipitation and temperature timeseries. The meteorological observations were extracted from the CLIMATO database system, managed by the Ministère de l'Environnement, de la Lutte contre les changements climatiques, de la Faune et des Parcs (MELCCFP,

2022), as well as from the Meteorological Service of Canada (MSC) Open Data Server, managed by Environment Canada and Climate Change (ECCC) (Montero et al., 2023). Meteorological forecasts corresponding to Regional Ensemble Prediction System (REPS) were retrieved from the Canadian Surface Prediction Archive (CaSPAr). The



variables of interest from the CaSPAr system correspond to precipitation at surface level and the temperature at 1.5 meters over surface (Montero et al., 2023). The minimum and maximum air temperatures required by the hydrological
modelling framework are derived from the probabilistic estimate of the ensemble forecast in a preprocessing step. The hydrometric dataset consists of station discharge and water levels timeseries along the study reach and was obtained from the Banque de données hydriques (BDH) database system, managed by MELCCFP (2022) and COBARIC. The water level records from BDH and COBARIC are in 15 minute and 1 minute interval, respectively.

The geographic data consists of shapefiles corresponding to the Chaudière basin, its sub-catchments, river and stream
network, and river gauging stations. This dataset was retrieved from MELCCFP (2020).

The different sources of data, as well as the period for which the data is available, are shown in Table 2.

**Table 2: Sources of data and periods of time for which data is available. Table adopted from Montero et al. (2023).**

| Data source | Variables | Period |
|---|---|---|
| BDH | Discharge | 2016-2023 |
| CLIMATO | Precipitation, Min, and Max Temperature | 2016-2023 |
| COBARIC | Water Level | 2019-2023 |
| CaSPAr | Precipitation and Mean Temperature | 2019 |
| ECCC | Precipitation | 2010-2023 |

In Québec, under-ice discharge estimates are produced by MELCCFP at a daily time step. This is done in accordance with the following internal procedures. Average daily discharge computed from the open water rating curve is corrected using a backwater correction factor. The MELCCFP has developed various methods to estimate backwater correction factor under dynamic river ice conditions to reduce the amount of uncertainty within its estimates (MELCCFP, 2019). For this study, the observed winter discharge was required at a finer resolution (hourly resolution).
Therefore, the uncorrected instantaneous discharge dataset was corrected by applying the backwater correction factor for the day. Figure 2 shows the application of this procedure at two stations, the Famine River (station ID 023422) and the Saint Lambert-de-Lauzon station (station ID 023402). It is interesting to note that the application of the backwater factor alters the shape of the resulting corrected hydrograph when compared with the uncorrected hydrograph. This could be due to the reason that the intraday discharge variation does not follow a fixed ratio but is
dynamic in nature while the backwater factor is calculated to estimate average discharge for the day and is thus a single value representing the backwater conditions at the site for that particular day. Furthermore, the backwater correction relies on a multiplicative factor, meaning that a smaller value of this factor represents larger backwater affect. In Figure 2, in case of the Famine River, the variation in backwater correction factor over the 5-day period is large i.e. from 0.3 to 0.5, whereas the variation at the Chaudière River at St. Lambert station is within a small range
i.e. 0.29 to 0.33. This indicates that the river ice conditions at the Famine River changed very quickly during this event. However, the overall trend in the variation of conditions remained the same at both stations.

It should be noted here that the under-ice river discharge estimates produced by MELCCFP using the method discussed above are uncertain and subjective to the skill of the operator in estimating the river ice conditions.



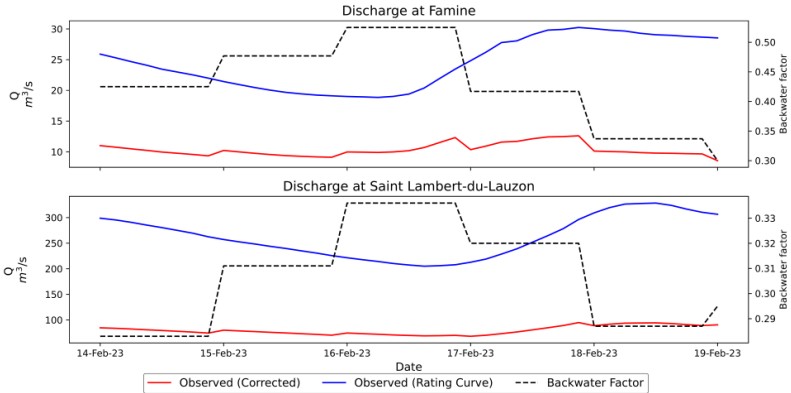


**Figure 2: Example of correction of instantaneous hydrograph using daily backwater flow factors calculated by MELCCFP for Famine and St. Lambert stations.**

## 3. Methodology

Hydrological models are commonly employed for streamflow forecasting and a well calibrated hydrological model

can provide relatively accurate streamflow forecasts under normal operational conditions. However, river ice conditions pose many challenges that cannot be incorporated into hydrological models such as hydraulic backwater effects of the ice cover's roughness, channel storage due to jamming, restricted flows, abstraction of water from the system owing to ice formation etc. Coupling a hydrological model with a hydraulic model offers a comprehensive modelling approach under river ice conditions. This framework provides a feedback mechanism between the natural

water cycle and the channel hydraulics, allowing the utilization of all data sources i.e. meteorology, hydrological states, and finally hydraulic controls (i.e. stage) to make fairly accurate discharge estimates and forecasts in near real time.

Operational forecasting systems deploy multiple tools and models in efforts to predict future events as accurately as possible.. Some of these tools are developed internally by forecasting agencies and are tailored to their particular needs

while others depend on open-source and commercial software (Montero et al., 2023). Regardless of the choice, coupling different tools and models is always a challenge for developers of such systems. The Delft-FEWS platform offers a flexible and configurable framework for storage and processing of hydro-meteorological data, as well as model integration and coupling hence, making it a simple platform choice for any forecasting system. The proposed forecasting system, as described in Montero et al. (2023), is conceptualized to force meteorological forecasts from

Environment Canada's Regional Ensemble Prediction System (REPS) onto the HydrOlOgical Prediction LAboratory (HOOPLA) toolbox to generate ensemble based hydrological responses from the study basin. The hydrological ensemble is then post processed and fed to an ice-cover unsteady hydraulic model developed in HEC-RAS, which produces a water level ensemble. The simulated water levels are then compared with the observed water level. The error magnitude guides the operator whether to accept the discharge estimates or readjust model parameters. The

different components of the forecasting system are described below.

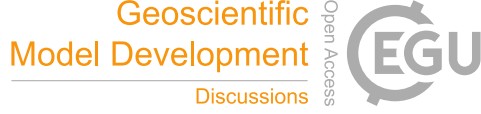

### 3.1 Regional Ensemble Prediction System (REPS)

The Regional Ensemble Prediction System (REPS), managed by Environment and Climate Change Canada, provides a probabilistic prediction of the different atmospheric elements over a 3-days forecast horizon. The REPS forecasts are generated by introducing small perturbations to the initial and boundary conditions of the model, which in return

create slight variations in the prediction (Montero et al., 2023). This effect was described by Lorenz (1963) as the butterfly effect, where small perturbations in the initial condition are propagated in a deterministic nonlinear model and produce large variations of the states in time (Montero et al., 2023). The REPS ensemble consists of 20 perturbed members as well as an unperturbed control member. Figure 3 shows an example of the REPS prediction for precipitation on the Chaudière River for 24 January 2019 (Montero et al., 2023).


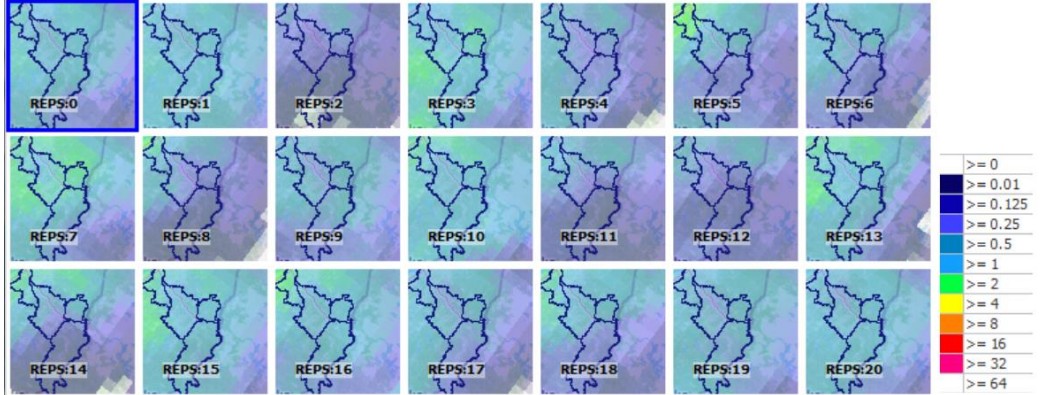

**Figure 3: Members of REPS precipitation (in mm) for the 24-01-2019. Figure adopted from Montero et al. (2023).**

### 3.2 Hydrological Module

The hydrologic module of the system is developed using the HydrOlOgical Prediction LAboratory (HOOPLA, version

1.0.1) framework, which is a multi-model hydrological modeling framework that consists of 20 conceptual lumped hydrological models, a snow accounting routine (SAR), data assimilation, and automatic calibration algorithms (Thiboult et al., 2019). A detailed description of HOOPLA can be found in Montero et al. (2023).

HOOPLA framework was calibrated and validated for the Chaudière system using the historically observed

meteorological and hydrometric timeseries from 2008 to 2018. The models in HOOPLA were calibrated automatically through the Shuffled Complex Evolution (SCE) (Duan et al., 1992) algorithm. The algorithms are iterative and global i.e., seek optimal parameter set within the parameter space (Montero et al., 2023). The MELCCFP provided gridded meteorological data at 0.1° – resolution grid, constructed through ordinary kriging from point observations obtained from a dense network of climatic stations across Québec (Usman et al., 2023). River discharge data was also obtained

for the same period and from the same source. During open water conditions the instantaneous discharge timeseries has 15 minutes resolution. However, during the river ice conditions the discharge data is produced at daily resolution and was therefore, downscaled by linear interpolation to 3h resolution to fit the model timestep requirements. The





entire dataset was split into two equal parts following KlemeŠ (1986) recommendations for model calibration and validation. Data from January 2008 to December 2012 was used for model calibration and from January 2014 to
December 2018 was used for validation. The year 2013 was used for model spin up before validation run. One of the salient features of the HOOPLA framework is that it is not data intensive and has rather simple data requirements. For calibration only precipitation, temperature, and discharge records are needed.

All 20 models included in HOOPLA framework were calibrated and validated for this study. For calibration, the computation timestep was 3h and snow accounting routine (SAR) was implemented. The SAR, Cemaneige, in the
HOOPLA framework spatially distributes the catchment into five elevation bands to compute snow accumulation and melt processes. The SAR is calibrated for each model individually (Thiboult et al., 2019). Figure 4 shows the performance of the modelling framework over the three study catchments for the 10-year long historical timeseries. Both calibration and validation periods show good agreement with the observations for all three catchments.

The modified Kling-Gupta efficiency (KGEm) was computed (Kling et al., 2012) as the calibration objective function
and same metric was used for evaluating model performance during the validation period. Figure 5 presents the distribution of the calculated KGEm for each catchment during model calibration and validation periods. The framework shows very high performance for all three catchments during the calibration period with no outliers and KGEm consistently above 0.88 (KGEm ranges between - ∞ to 1, with 1 being perfect fit). During the validation period the Intermediate Chaudière catchment showed nearly similar performance to its calibration with KGEm ranging
between 0.89 to 0.93 for all the models. In the case of Famine catchment, the HOOPLA framework performance during validation was slightly superior to the calibration period with KGEm values ranging between 0.89 to 0.92 for most of the models except for one outlier, Model 8, which had a KGEm value of 0.83. In the case of Upper Chaudière catchment, the performance during validation was inferior to calibration, for this catchment the performance metric remained in a range of 0.84 to 0.90 during the validation period which is still considered as very good performance
by the framework.

### 3.3 Hydraulic Module

The next module in the coupled modelling system is the hydraulic module developed in HEC-RAS (version 6.0) (Brunner, 2016). This module consists of two hydraulic models: i) a 1D unsteady river ice model to simulate simple ice cover hydraulics, and ii) a 1D steady state river ice model to simulate ice jam profiles for the Chaudière River. The
1D steady state model is not used in the current study and is therefore, not described here. The modelled reach is located entirely within the Intermediate Chaudière catchment. The length of the modelled reach is 81.18 km (from chainage 106.78 to 25.60 km). The models have upstream boundary condition defined downstream of the Sartigan Dam and the downstream boundary condition defined at the hydrometric station at Saint Lambert-de-Lauzon (CEHQ station 023402).



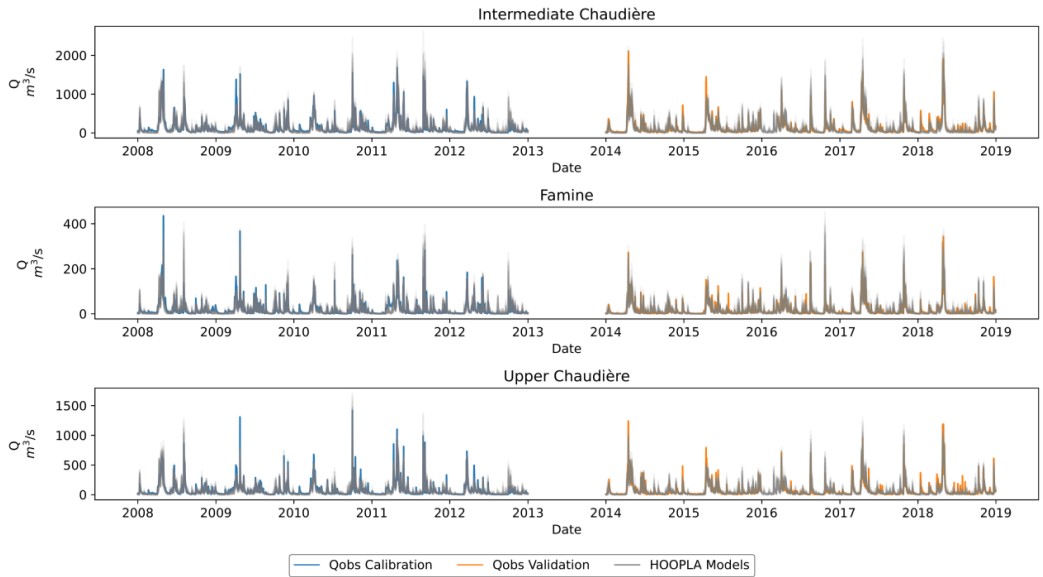


**Figure 4: Calibration and validation of HOOPLA framework performed on the three sub-catchments defined in the Chaudière River Basin.**

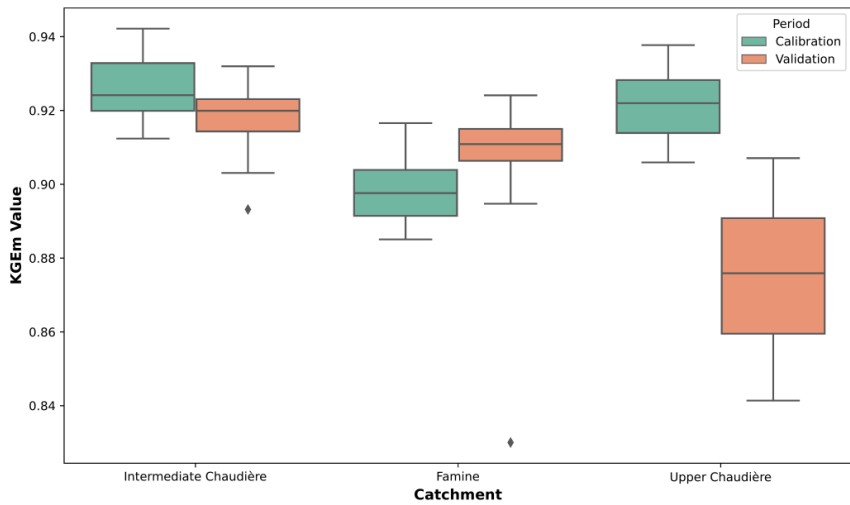

**Figure 5: Boxplot showing the modified Kling-Gupta Efficiency (KGEm) distribution of all 20 hydrological models in HOOPLA framework for each catchment during calibration and validation periods.**

The hydraulic model was adopted from the work of Ladouceur (2021) where a 1D steady state river ice model for the Intermediate Chaudière reach was developed to simulate ice jams and determine flood levels. The model geometry

was constructed by merging a digital terrain model (DTM) built from LiDAR survey carried out by Ministère des



Ressources naturelles et des Forêts (MRNF) and bathymetric surveys conducted by MELCCFP in 2005 and University Laval team in 2020-21. The model geometry consisted of 481 cross-sections and includes most of the bridges within the reach. The details of these surveys can be found in Ladouceur (2021).

The steady state river ice hydraulic model by Ladouceur (2021) was modified and converted to 1D unsteady river ice model. To adapt the model for low flows more cross-sections were added through cross-section interpolation in HEC-RAS, cross-section density was especially increased near hydraulic structures and locations along the river reach where bed slope was changing. Figure 6 shows the schematic of the 1D unsteady river ice hydraulic model used in the study. The upstream boundary condition is defined as an inflow hydrograph from the Upper Chaudière, introduced downstream of the Sartigan Dam station (Station ID: 023429) at the first cross section of the model (chainage 106.78 km). A lateral inflow hydrograph, representing the flow from the Famine River (Station ID: 023422), is introduced at chainage 101.37 km. The downstream boundary condition is set as normal depth conforming to uniform flow conditions at Saint Lambert-de-Lauzon (Station ID: 023402, chainage 25.60 km).

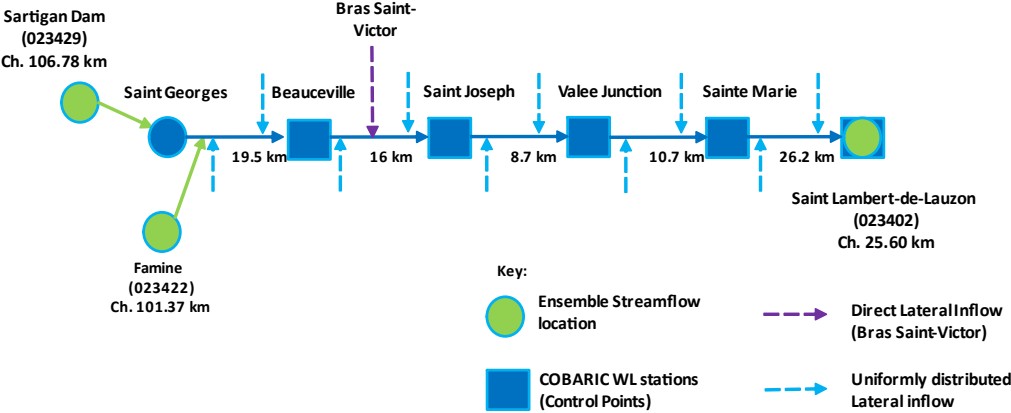

**Figure 6: Schematic of the 1D unsteady river ice Hydraulic model setup in HEC-RAS for the Intermediate Chaudière reach. The green circles represent ensemble streamflow locations obtained from the hydrological module of the system. The blue squares represent the water level control points located along the reach.**

The Intermediate Chaudière catchment has several small sub-catchments. The challenge with these sub-catchments is that they are mostly ungauged. The total area of these sub-catchments is approximately 2100 km$^2$, with individual catchment sizes ranging from 20 to 720 km$^2$, which is considerable. Therefore, the lateral inflow contribution of these sub-catchments cannot be ignored. Bessar (2021) developed a very simple volume-based approach to derive ungauged lateral flows for the Chaudière River. This approach calculates a difference hydrograph (mathematically expressed in Eq. (1)) from the routed upstream hydrographs (i.e. the hydrographs at Upper Chaudière and Famine routed through the hydraulic model) and the observed or modelled (in case of forecast) downstream hydrograph at the model outlet (i.e. Intermediate Chaudière hydrograph at Saint Lambert-de-Lauzon station). This difference hydrograph is then split into two hydrographs: a uniformly distributed lateral inflow hydrograph ($Q_{ULI}$ in Eq. (2)) which amounts to 60% of the calculated difference hydrograph, and a direct lateral inflow hydrograph, representing the Bras Saint-Victor



tributary ($Q_{BSV}$ in Eq. (3)), which is 40% of the calculated difference hydrograph. The uniformly distributed lateral

inflow hydrograph accounts for the inflow from most of the small sub-catchments while the direct lateral inflow hydrograph accounts for the Bras-Saint Victor tributary of the Chaudière River, which has an area of 720 km$^2$ and is approximately the size of Famine catchment (Figure 6).

$$Q_{ungauged} = Q_{IC\ Observed/Modelled} - Q_{IC\ Routed} \tag{1}$$

$$Q_{ULI} = 0.60 * Q_{ungauged} \tag{2}$$

$$Q_{BSV} = 0.40 * Q_{ungauged} \tag{3}$$

Where $Q_{ungauged}$ is the discharge from the ungauged catchments, $Q_{IC}$ is the outflow hydrograph (observed or modeled) of the Intermediate Chaudière at Saint Lambert-de-Lauzon station, and $Q_{IC\ Routed}$, is the inflow hydrograph routed

through entire reach. It was estimated that a 12-hour delay in the application of the lateral inflow hydrographs is appropriate based on the examination of several winter hydrographs. This approach, however, introduces a new source of uncertainty in the modelling chain with regards to the application of the ungauged inflow hydrographs.

The calibration of the model using the water levels from the COBARIC hydrometric for open water conditions was done for bank full discharge conditions. This required the discharge to be in a range of 700 to 1000 m$^3$/s. The

calibration showed that water level errors were within $\pm10$ cm and the peak discharge simulated was 1002 m$^3$/s observed at Saint Lambert-de-Lauzon station (Ladouceur et al., 2023). After performing open water calibration, the model was calibrated under ice cover conditions. River ice data including ice thickness and average daily flows were collected by Laval University during the winter of 2019 and 2020. This data was used for calibrating and validating the hydraulic model. Calibration was done on the data collected in 2020, the average discharge was 9.15 m$^3$/s and ice

thickness along the reach ranged from 0.4 m to 1.05 m. The model showed an RMSE of 0.25 m in the calibration phase. Validation was performed on 2019 dataset, where the average daily discharge was 19.33 m$^3$/s and ice thickness varied from 0.35 m to 0.74 m along the reach. The RMSE calculated during the validation phase was 0.37. Further details on calibration and validation can be found in Ladouceur et al. (2023).

### 3.4 Delft-FEWS testbed framework

Delft-FEWS (release 2022.02) is software platform that enables the configuration of operational forecasting systems in a flexible manner integrating existing models and available data (Werner, Schellekens et al. 2013). In contrast to many other forecasting systems, it contains no inherent hydrological modelling capabilities within its code base. Instead, it relies entirely on the integration of external (third party) modelling components (Montero et al., 2023). Delft-FEWS is extensively used in Canada by different local and government agencies, among them are: Alberta's

River Forecasting Centre, the Water Security Agency in Saskatchewan, the MELCCFP of Québec, New Brunswick's River forecast Centre, the territorial governments in the Yukon and Northwest Territories, along with key utility providers such as BC Hydro, Manitoba Hydro, and Ontario Power Generation (Arnal et al., 2023; Montero et al., 2023).

The Delft-FEWS platform can be considered as a model-agnostic platform (Arnal et al., 2023), which takes care of

preparing the data that each individual model requires to produce a simulation. This is done through the concept of a



"General Adapter", which is a processing module in Delft-FEWS that allows to connect the data to the model by three basic components: i) exporting data from the system to specific directories as published interface-extensible markup language (pi-xml) or NetCDF files, ii) executing activities, and iii) importing data back into the system from pi-xml or NetCDF files. More specifically, the first component consists of exporting model states, model parameters, model

setup, as well as the time series data required for the model run. The second component is a model specific one, which interacts with a model adapter. The model adapter is split into three basic routines, the first one being the pre-adapter that transforms the pi-xml data into the native model specific format (pre-adapter), the second one executes the program (adapter), and the third one transforms the native model specific data back into pi-xml files (post-adapter). The third component of the General Adapter reads the model states as well as the time series data and log messages

back into the Delft-FEWS database.

Delft-FEWS also includes additional modules designed to support the forecasting processes, among them: import routines, transformation processes, and export routines. These modules allow to produce the operational data necessary to run external model simulations as well as to generate relevant data for the operators of the system. It also allows to create information in multiple forms, readily available to be used for decision-making processes (e.g. reports,

tables, graphs). Furthermore, there are extensive data visualisation options through different types of displays, there is advanced data archiving tailored to storage of forecast data and related products, and it contains a training and exercise module. These modules are all connected to a centralized database, which keeps track of the data rather than a specific model, and ensures all forecasters and end-users inspect the same data. This is described by Gijsbers, et al., (2008) as a data-centric approach rather a model-centric approach. The main advantage of a data-centric approach

relies on the flexibility of the system, as it provides a "shell through which an operational forecasting application can be developed specific to the requirements of an operational forecasting centre" (Werner et al., 2013).

The testbed system for the Chaudière River enables the integration of: i) observed and prognostic meteorological variables, ii) hydrological modelling toolbox, iii) hydraulic model component, as well as components that enable iv) real-time modification of model parameter and data processing. Figure 7 illustrates the process of the forecast

workflow and the main variables that are involved in each step of the forecast (Montero et al., 2023).

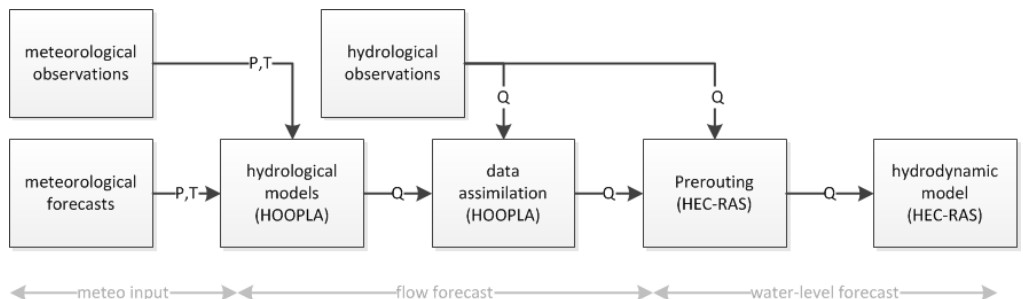

**Figure 7: Schematic overview of workflow for the hydrological forecasts. P is precipitation, T is air temperature, Q is discharge. Figure adapted from Montero et al. (2023).**



The workflow manages the meteorological and hydrological observations, as well as the meteorological forecasts. The observations are pre-processed in the platform to calculate basin average values using Kriging interpolation that are required by the hydrological toolbox (HOOPLA) to simulate the discharge data to be used in the hydrodynamic model (Montero et al., 2023). Finally, the hydrodynamic model is compared to observations to adjust any parameters or

variables within the workflow (Figure 7). Note that HOOPLA assimilates the hydrological observations to produce updated model states to improve the hydrological simulation. The system is configured to produce a pre-routing step that allows to create simulated hydrological inflows to the unsteady hydrodynamic model of the Hydraulic Module. The processing diagram to create the basin average variables is illustrated in Figure 8 (Montero et al., 2023). The processing for precipitation is done by merging together individual station observations to create a spatially distributed

precipitation grid over the catchment through the Kriging interpolation procedure . The spatially distributed precipitation can then be averaged within each basin (Montero et al., 2023). The same procedure is performed for the observed temperature variable, with an additional step of estimating the temperature of each station at sea level. After which the Kriging interpolation and the subsequent basin averaging procedures are performed (Montero et al., 2023).

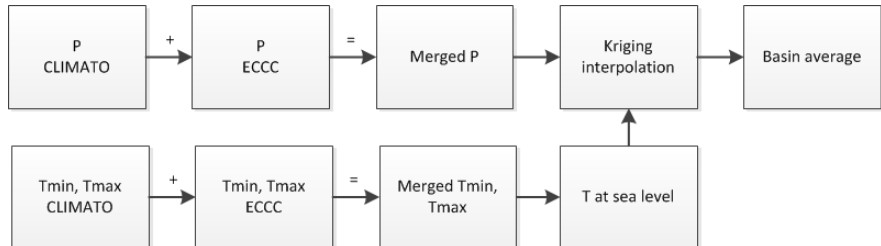


**Figure 8: Preprocessing of observed data to create inputs for hydrological historical update runs. P is precipitation and T is air temperature. Figure adopted from Montero et al. (2023).**

From an operational perspective, it is desirable to be able to interact with the models and to be able to modify sensitive

parameters in a convenient manner. This flexibility in Delft-FEWS is provided by *Modifiers*. The current system for the Chaudière River is configured to have three modifiers' groups (Montero et al., 2023). The first modifier corresponds to the modification of the parameters used as a proxy of the hydrological routing to compute the ungauged inflows between Sartigan and Saint Lambert-de-Lauzon stations (Montero et al., 2023). The parameters corresponding to this first group of modifiers are the shift in the hydrograph (i.e. the time lag in the application of the

difference hydrograph) and the split factor i.e. the ratio in which the difference hydrograph should be divided into to estimate the uniform and direct lateral inflow as described in **Section 3.3**. By default, the hydrograph shift is kept to 7 hours in the Delft-FEWS configuration whereas for the distribution of the difference hydrograph (i.e. $Q_{ungauged}$ in Equation 1) the ratio is set as 60:40 between the uniformly distributed lateral inflow and the direct lateral inflow.

The second modifier group allows the operator to modify the ice cover in the HEC-RAS model (Montero et al., 2023).

These modifications correspond to ice thickness and under-ice roughness. Following the river ice characterization described in Ghobrial et al. (2023), the model was split into 9 reaches (subject to alteration). The ice conditions were





considered homogenous within each reach. The modifier has these pre-defined reaches, where the ice thickness and ice roughness can be changed both for the main channel as well as for the banks (Montero et al., 2023).

The third modifier group allows the operator to directly modify the inflow hydrographs (through operations such as
multiply, divide, add or subtract by certain factor) that are linked to the boundary conditions of the hydraulic model, i.e., the hydrographs at Sartigan, Famine, Saint Lambert-de-Lauzon, as well as the hydrographs at Bras-Saint-Victor and the uniformly distributed flow (Montero et al., 2023).

Figure 9 depicts the main components of the user interface of the Delft-FEWS River Ice Testbed system for the Chaudière River. Note that the platform provides diverse information regarding spatial distribution of meteorological
variables (as seen in Spatial Data Display), the visualization of real-time scalar information showing observed and simulated time series at specific gauging stations (through the Hydrological Module Display), as well as the three possible modifiers to introduce operational decisions to the forecast workflow (via Modifiers Configuration). The platform is intended to facilitate the complex dynamics of river ice modelling by updating hydrological and hydraulic components and a quick verification with observed time series imported into the system. The complete forecast
workflow is summarized in the Forecast Tree, where a series of nodes trigger individual components of the workflow such as the meteorological data imports, hydrologic simulations, and the hydraulic simulations, both for the update and the forecast runs.

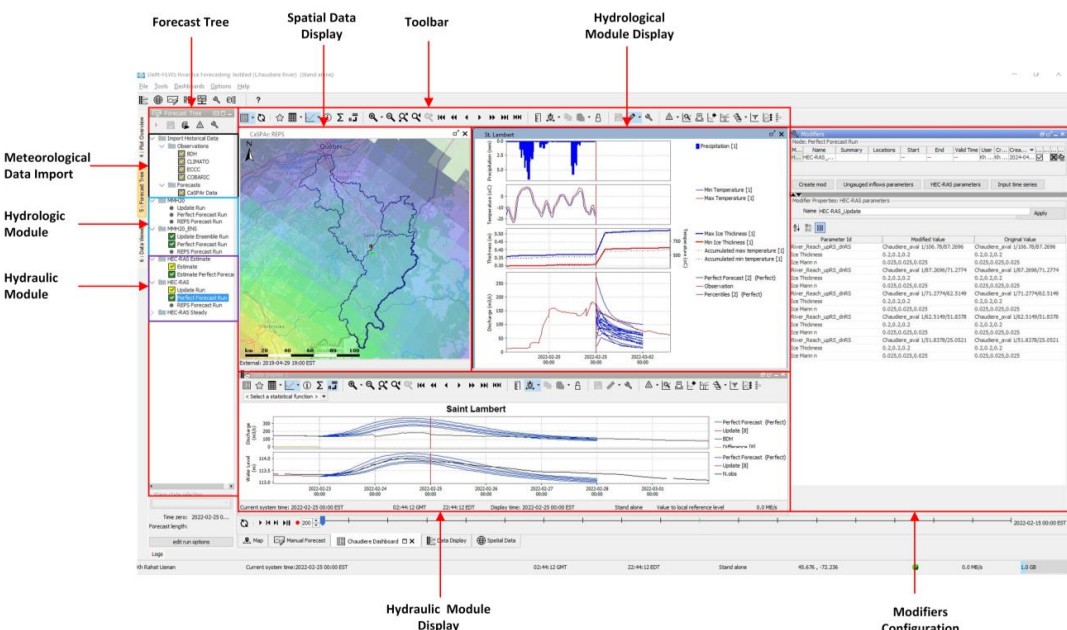

**Figure 9: A screenshot of the Delft-FEWS River Ice Forecasting testbed system for the Chaudière River.**

The system has been developed for operational forecasting of under-ice river discharge in a probabilistic manner. The probabilistic approach helps address various sources of uncertainty in the modelling chain such as: meteorological





forcings uncertainty; hydrological model structure uncertainty; model states uncertainty; and parametric uncertainty,

as well as captures variability and presents a range of future scenarios to the decision makers who can then make better informed decision. However, an important aspect of an operational forecasting system is the computation time. Ensemble based methods are usually computationally expensive and time consuming. For example, with a 2.90 GHz 8-Core(s) processor and 16 GB of RAM, a single forecast run for HOOPLA, consisting of only one meteorological member for the three catchments takes approximately 2 minutes and a single member unsteady run in HEC-RAS takes

approximately 4 minutes. From this we can estimate that a 20-member ensemble run for a single forecast in HEC-RAS will take approximately one hour for a 5-days long forecast. This can be solved in Delft-FEWS by splitting the workflow into multiple computational nodes that run in parallel, if necessary.

Both the Hydrological and Hydraulic modules are configured to have an "**Update Run"** component in addition to the "**Forecast Run**" component. The function of the Update Run component in both these modules is to produce updated

model states for a Forecast Run. The Update Run feeds on available observed data prior to the forecast issue time (i.e. $T_0$). The simulation performed by the models in this run create a new set of model states (i.e. updating storages in hydrological models, and water levels in the hydraulic model) that best describe the catchment conditions. In case of the hydrologic module, data assimilation is implemented in the Update Run component through the Ensemble Kalman Filter (EnKF).

The hydrological modelling framework HOOPLA can take both deterministic and ensemble meteorological forecasts as input and produce an ensemble output. The output ensemble size depends on the size of meteorological ensemble, the number of models selected for forecast run from the framework and the number of perturbations in the data assimilation procedure. For example, if all 20 models of the hydrological modelling framework are forced with a REPS forecast which consists of 20 meteorological members, and data assimilation is applied by perturbing 50

members, then the resulting forecast ensemble has a size of 20,000 members (20x20x50). Processing all these members through the hydraulic module of the system will be impractical in an operational context. Therefore, the output from each hydrological model is reduced to a single member which represents the average of the forecast ensemble for that model and thereby, leaving us with a 20-member hydrological forecast ensemble.

The hydrological forecast ensemble is further post-processed to calculate ensemble statistics (i.e. percentiles). The

20th, 33rd, 50th, 66th, and 80th percentiles of the resulting ensemble were calculated. These five members are fed to the hydraulic module to simulate water levels.

### 3.5 System Testing and Evaluation

For the current study, the system was tested by hindcasting events for which manual discharge and ice thickness measurements were available. This is important since the inherent uncertainty within the winter records is significantly

large. Moreover, to reduce the element of meteorological uncertainty in the hindcasted events, the observed meteorological data was used as forcing data which is termed as **"Perfect Forecast."** For this study, the hindcasted events are summarized in Table 3. For each of these events, the forecast horizon is set as 5 days for both the hydrological and hydraulic module. However, it should be noted that the manually measured discharge is available only at the first day of each forecast and for the remaining length of the forecasted period the backwater corrected



discharges were used for comparison between observed and simulated flows. Also, the measured discharges for the Intermediate Chaudière (measured at Saint Lambert-de-Lauzon gauging station) are used, whereas for the Upper Chaudière and Famine sub-catchments backwater corrected discharges are used for comparison with simulations. This is done with two main considerations: (i) Intermediate Chaudière is significant for lateral inflow estimation, and (ii) Intermediate Chaudière hydrometric station also produces stage data which is used in the evaluation of the hydraulic module of the system, hence, accurate discharge information at this location is critical for performance evaluation of the system.

**Table 3 Summary of the Events used to evaluate the system.**

| Event | Start Date | Accumulated Precipitation (mm) | Average Daily Temperature (°C) | Discharge at Saint Lambert-de-Lauzon (Station ID 023402) (m³/s) | Ice thickness (m) |
|---|---|---|---|---|---|
| 1 | 05-02-2020 | 0.7 | -7.3 | 24.80 | 0.30 |
| 2 | 23-02-2022 | 6.45 | 1.2 | 138.30 | 0.50 |
| 3 | 14-02-2023 | 0.37 | -0.5 | 55.60 | 0.28 |
| 4 | 28-02-2023 | 7.2 | -5.8 | 25.10 | 0.34 |

For the first event (in Table 3) detailed ice thickness measurements along the modelled reach were available from Ghobrial et al. (2023) and were used in the model. For the remaining events a measured ice thickness at key locations along the river was used. The performance of the hydrologic module is evaluated at all three control stations i.e. the hydrometric stations. While for the hydraulic module the performance is evaluated at two water levels stations which are Sainte Marie and Saint Lambert-de-Lauzon. There are two reasons behind excluding the rest of the COBARIC stations. First is the reliability of stage data which is the case at Beauceville where the station is location under a bridge and immediately downstream of an island, thus the ice conditions at this location are affecting the local stage in an unexpected manner which cannot be accounted for in a simple hydraulic model (Ladouceur, 2021). Second, the sensor at some gauging stations is not installed at the thalweg of the river, and thus cannot measure water levels below a specific threshold. This is the case for the Valee Junction station where there is a threshold elevation of 142.96 m, below which the station cannot produce water level records.

The preliminary evaluation of the system is conducted by computing root mean square error (RMSE) and percent bias (Pbias) as deterministic scores. RMSE is a commonly used metric which is a measure of average magnitude of error between model predictions and observations. A lower value of RMSE is an indicator of good model performance. However, this metric is absolute metric and does not indicate the direction of error i.e. either the model is overestimating or underestimating. For determining the direction of error, the second metric which is the percent bias (Pbias) is used. Pbias is a metric that determines the error in model prediction relative to the observations and indicates the direction of error i.e. positive error (overestimation) or negative error (underestimation). A value of Pbias close to zero is desirable to have an unbiased modelling system. The deterministic metric is calculated from the average ensemble values for each event (Anctil and Ramos, 2017). Since the current dataset is comprised of only 4 events,



bootstrapping was applied to calculate the 90% confidence intervals for the deterministic scores for a detailed
description of system's performance (Velázquez et al., 2010; Bessar, 2021).

## 4. Results and Discussion

Before discussing the performance of the coupled modelling system through performance metrics, a single hindcast
event is presented to provide an overview of the systems capabilities in an operational context.

### 4.1 Hindcast February 23-25, 2022

A single streamflow and water level hindcasting event is presented in Fig. 11 and 12, respectively, to explain the
everyday operational procedure of the system for forecasts issued on three consecutive days from February 23 to 25,
2022. MELCCFP carried out discharge measurement at the Intermediate Chaudière hydrometric station (Station:
023402) on February 23, 2022. The measured discharge at this location averaged for the day was reported to be 138.30
$m^3$/s. The ice thickness measured by the Laval University team on February 22, 2022, at Chaudière on average was
0.5 m. For the Upper Chaudière and Famine sub-catchments corrected winter discharge is considered since measured
discharge is not available on the same day.

Meteorological observations were used as forcing data (Perfect Forecast) to keep the forcing uncertainty low. Figure
10 shows the meteorological conditions observed during the forecasted period. The accumulated precipitation between
February 23 to March $2^{nd}$, 2022, was recorded to be 22.1 mm for the Upper Chaudière, 18.9 mm for the Famine
watershed and 20.3 mm for the Intermediate Chaudière. The temperatures during this period mostly remained below
freezing indicating that most of the precipitation received was solid precipitation (i.e. snow). However, positive
temperatures were recorded for the three sub-basins on February 23, 2022; the precipitation data shows a rainfall event
making the conditions suitable for runoff generation.

The hydrologic module of the couple modelling system was updated for the hydrologic states by running it in a batch
simulation mode from November 15, 2021, till February 23, 2023, at a daily interval (this is not to be confused with
computation time step of the hydrological modelling framework which is 3-hours). This means that for a duration of
100 days, the hydrological model states were updated every day through continuous simulations and data assimilation.
This sets up the model states in optimal conditions for forecasting. Perfect forecasts for February 23, 24, and 25 were
run (each forecast run is followed by an update run to set up the hydrologic states for the next forecast run). After the
hydrologic module has finished running, the hydraulic module of the system is executed in a similar manner. The
hydraulic module is first run in update mode where the input data consists of observed flows. This run updates the
states parameters of the hydraulic module based on observations. This allows the operator to make a visual comparison
between the simulated an observed stage at $T_0$ ($T_0$ is the time at which the forecast is to be made) and update/modify
any required model parameter such as ice thickness or under ice roughness. The next step is to run the hydraulic
module in forecast mode where the input data is now passed from the hydrologic module consisting of the forecasted
streamflow ensemble.



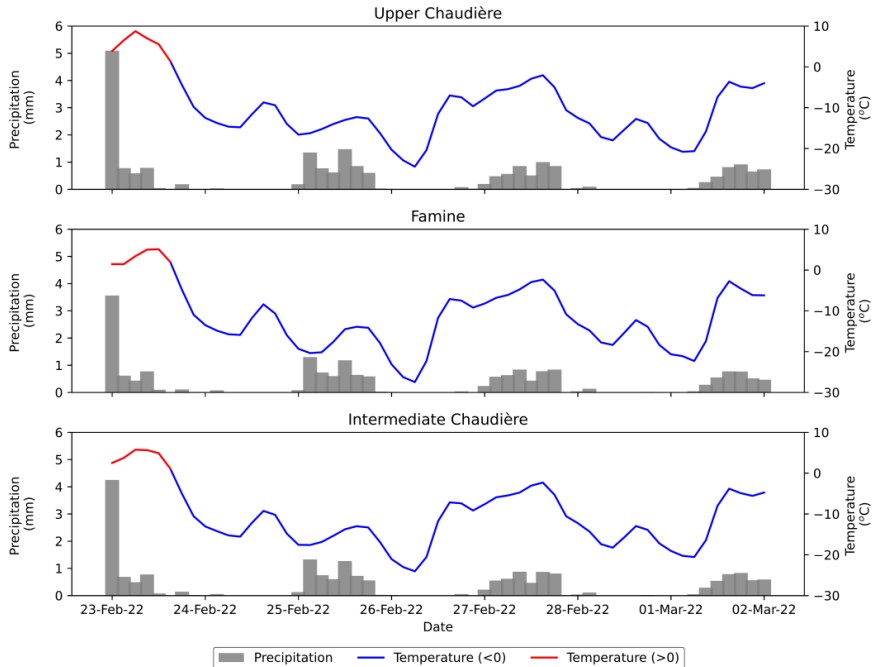

**Figure 10: Meteorological observations used as forcing data ("Perfect forecast") for the hydrologic module to simulate the event. The average temperatures during the forecast period remained below freezing.**

Figure 11 presents the hydrological forecast for the three modelled catchments. Compared to the corrected observations produced by MELCC, the forecast issued on February 23, 2022, shows that the models overestimated the streamflow flow for the Upper Chaudière sub-catchment. For Famine the observed hydrograph falls within the ensemble for the first few timesteps (8 timesteps) and later the models tend to underestimate the discharge. For the Intermediate Chaudière there is overestimation in the peak discharge, but the hydrograph recession mostly falls within the ensemble bounds. Subsequently, hydrological forecasts are issued on 24th and 25th February 2022. For the Upper Chaudière, the gap between the observed and forecasted hydrograph is now reduced and in fact the observations fall within the ensemble bounds (the ensemble is quite narrow in this case) until the March 01, 2022, after which the models underestimate the flow. For Famine, the forecasted hydrograph is an underestimated one. The impact of data assimilation and model states update is visible here, for the subsequent forecasts the discharge at the forecast issue time (i.e. 24th Feb and 25th Feb respectively) is higher than the previously forecasted values however, it is still not high enough to match the observations. In the case of Intermediate Chaudière, the error at forecast issue time is high but starts reducing over time, as the forecast horizon is reached the forecasted hydrograph dips below the observed one and the observations no longer fall within the ensemble bounds.



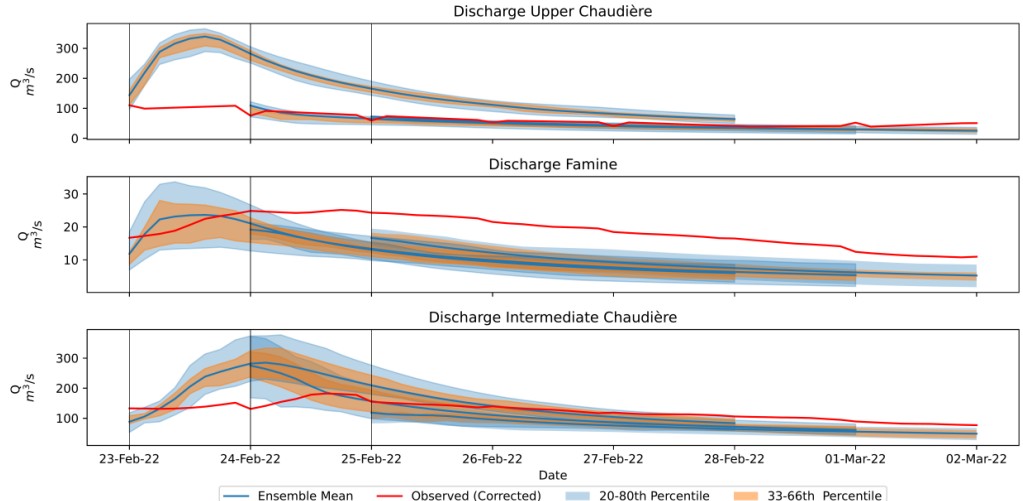

**Figure 11: Ensemble hydrologic forecast evolution for February 23, 24 and 25, 2022 at the three sub-catchments of the Chaudière River. The black vertical line in the plot corresponds to the issue time of each forecast.**


Figure 12 presents the results from the hydraulic module of the system. For the water level forecast issued on February 23, 2022, the observed water levels at Sainte Marie fall within the ensemble for the first four days of the forecast period and on the fifth day the observed levels are higher than the simulated ones. The ensemble mean remain higher than the observed levels for the first three days and later dips below the observed levels. The result at Saint Lambert-

de-Lauzon station for the forecast on February 23 is similar to Sainte Marie, however the difference is that the ensemble mean and observed water levels are superimposed indicating a very small error between modelled and observed levels for the first three days but later the forecast ensemble dipped below the observations. For the subsequent forecasts i.e., forecasts issued on 24th and 25th February the error in water levels is higher. Montero et al. (2023) have demonstrated the sensitivity of water levels to discharge. The fact that the simulated water levels matched

the magnitude and shape of the observed water levels (Fig. 12) improves confidence in the simulation and indicates that the system is performing very well when using the simulated inflow hydrographs from HOOPLA as input to the HEC-RAS model. The fact that these inflow hydrographs were much higher than the discharges reported by MELCCFP (Fig. 11) indicate that the reported under ice discharges may not be accurate and the uncertainty with regards to the estimation of backwater factor used to produce these discharges is very high.

The measurement done by MELCCFP on 23rd February 2022 is a point measurement, the intraday discharge variability cannot be established from a point measurement, furthermore, the discharge estimates produced are reported as an average for the entire day hence the intraday variability is lost. Hydrological modelling produces discharge estimates at a finer resolution (i.e. 3-hour resolution) and the water level records, which are available at an even finer resolution (i.e. 1-minute resolution), can be used to confirm streamflow projections through hydraulic modelling.




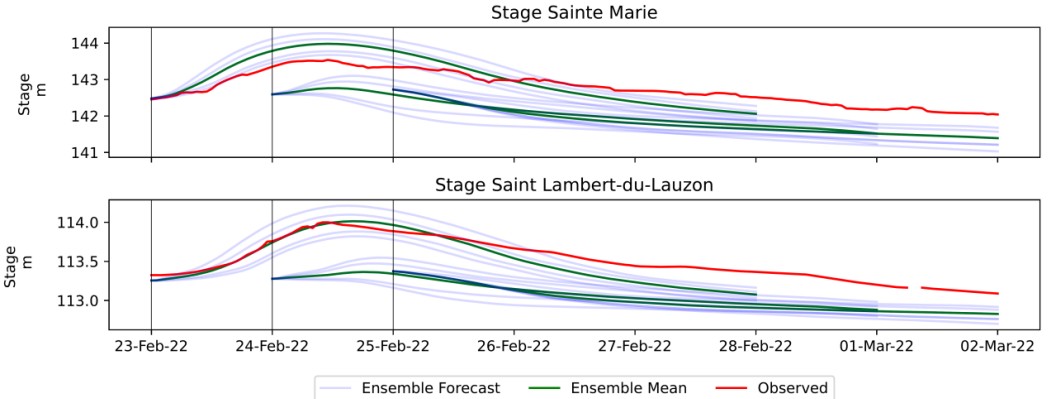

**Figure 12  Ensemble water level forecast at the two water level stations along the Chaudière River for forecasting on February 23, 24 and 25, 2022. The black vertical line in the plot corresponds to the issue time of each forecast.**

### 4.2 System Performance

The performance metrics evaluated for the system are presented in Figure 13. The top panels represent the results for the hydrologic module and the bottom panels represent the results for the hydraulic module. The metrics are evaluated over the forecast lead times. Since the system ran in "Perfect Forecast" mode, the meteorological uncertainty is low. The RMSE evaluation of the hydrologic module shows stable performance over the first two days of the forecast window for all three sub-catchments. In this period the RMSE is small, and the confidence intervals are narrow.

However, for the Upper and Intermediate Chaudière catchments the RMSE starts increasing sharply from the T+2 and reaches a peak value nearly 3.5 days from $T_0$. The Upper and Intermediate Chaudière sub-catchments show similar trend. Average daily discharge during the winter period was calculated for each sub-catchment over a period of seven winters between 2017 to 2023 for comparison with RMSE. The magnitude of RMSE for Upper Chaudière is in 8-37 m$^3$/s range, this value is smaller as compared to the average daily discharge of the Upper Chaudière which was found

to be 85.5 m$^3$/s. At Intermediate Chaudière the RMSE is in 20-42 m$^3$/s range over the forecast horizon while the average daily discharge was found to be 165.5 m$^3$/s. For Famine the RMSE remains stable throughout the forecast and operates within a narrow range. For Famine the RMSE range from 3-4 m$^3$/s over the forecast window while the average daily flow for Famine was found to be 20 m$^3$/s.

The Pbias analysis for the hydrologic module suggests that the module underestimates discharge for all three sub-

catchments in the first two days of the forecast window. The underestimation continues for the Intermediate Chaudière and Famine sub-catchments. However, the Upper Chaudière sub-catchment is overestimated after T+2 days. The Intermediate Chaudière sub-catchment starts with 20% underestimation of the discharge that fluctuates over the lead time. The Pbias for this sub-catchment remains in a range of -27% to 3.5%. For Upper Chaudière the Pbias ranges from -24% to 72.1%. This is an extremely high variation and indicates an erratic behavior in hydrological forecasting

for this catchment, however it is important to keep in mind the uncertainty within the winter records and further loss in documenting the intraday variability. For Famine, the forecasted discharge remains underestimated and the Pbias values remain a range of -45.15% to -23.33%.



The hydraulic module is also evaluated with the same metrics. RMSE is calculated for Sainte Marie and St Lambert stations. The RMSE at the two water level stations is in rising trend while moving forward in the forecast horizon. For

Sainte Marie the RMSE is in a range of 10 to 50 cm. The rise in error is relatively sharp in the initial forecast periods and the error peaks at T+4 days. For St Lambert station, the RMSE increase rather gradually and is in a range of 10 to 20 cm. The confidence intervals are narrow.

The Pbias analysis of the water levels reveals an overestimation of water levels for Sainte Marie station in the first two days of the issued forecast, followed by underestimation for the remaining period. The Pbias at this station ranges

from -10% to 8% however, the confidence intervals for Sainte Marie station are wide indicating higher uncertainty at this station. For St Lambert station the Pbias performance is quite stable as was the RMSE. The analysis shows an underestimation of water levels at this station however, the Pbias varies in a range of -12% to -4%. The confidence intervals estimated for this station are quite narrow.

The current analysis of the coupled modelling system is done with a limited dataset which renders it sensitive to

extreme events. The analysis for the hydrologic module shows higher deviations from the observations as quantified through the Pbias metric especially, for the Upper Chaudière sub-catchment. Similarly, Pbias is also in a higher range for Famine catchment and shows significant underestimation. Given the inherent uncertainty in the observed data it is difficult to identify the main cause behind this behavior. The hydraulic module on the other hands shows satisfactory performance. The Pbias and RMSE values are within narrow and acceptable range. This eventually casts doubts on

the quality of observed discharge data used for comparison as water levels are more sensitive to discharge than any other parameter.

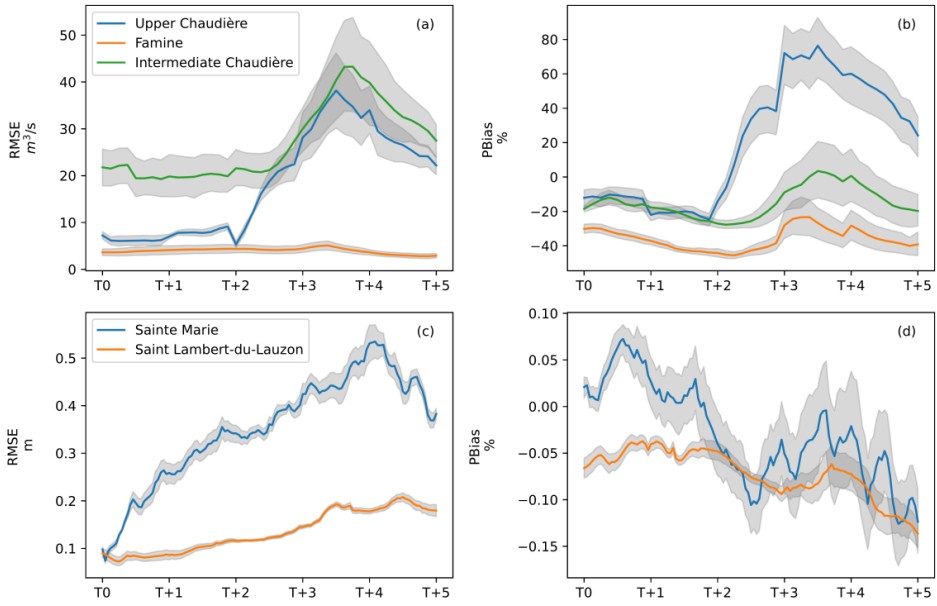

**Figure 13: Performance evaluation of the coupled modelling system using RMSE (subplots a and c) and Pbias (subplots b and d) scores. The top panel represents the results of hydrologic module, while the bottom panel represents the results of**
**hydraulic module.**



### 5. Conclusions

In this study we demonstrate the development and capabilities of a coupled (hydrologic and hydraulic) modelling system for operational forecasting under river ice conditions and assess the system's functionalities over selected winter events. The system is developed for the Chaudière River in Québec, Canada. The hydrologic modelling framework HOOPLA was coupled with a 1D unsteady river ice model in HEC-RAS. The operational forecasting system was configured in Delft-FEWS, which provides a flexible environment for data management and model integration.

The current study conducted a preliminary analysis of the developed system based on handful events where in-situ data was available. The under-ice river discharge data retrieved from observations is highly uncertain since there is no reliable method available till date for its estimation other than direct measurements, which are usually sparse given challenging conditions. An additional challenge with this data is its resolution. Winter records are usually produced at a daily resolution which fails to account for the intraday variability, although it can be argued that the winter period is usually stale with not much flow variability within a day however, considering the dynamic nature of river ice processes, especially during the breakup period, a finer resolution of under ice discharge is of value. The chosen events were simulated using meteorological observations as forcing data (i.e. **Perfect forecast**) and were analyzed using deterministic scores namely RMSE and Pbias. The hydrologic module of the system under-estimates winter flows for two sub-catchments and significantly overestimated flow for Upper Chaudière sub-catchment. Despite the RMSE values being low, the simulations showed significant deviations from the observed (corrected rating curve discharge) (i.e. Pbias metric).

The hydraulic module showed better performance with the RMSE staying within an acceptable range in the initial forecast lead times. The water level forecast generally remained under-estimated as evident from the Pbias analysis, but it was within an accepted margin. This analysis though needs to be interpreted with caution since it is based on limited data and warrants further investigation using a larger dataset to improve confidence on the system.

The current study lays foundations for a modelling system that can be applied for reliable estimation and forecasting of winter flows. The coupled modelling approach has demonstrated potential in resolving the long-standing challenge in the estimation and forecasting of under-ice river discharge. This approach provides a comprehensive method to the forecasting agencies in cold climate region to manage water resources throughout the year and reduces the element of subjectivity in the under-ice discharge estimates. It also provides a mechanism to make use of all available hydrometeorological variables such as precipitation, temperature, streamflow, and water level information to estimate under-ice flows at desired temporal resolution. The ensemble-based approach caters for the uncertainty arising from various sources in the modelling chain and presents a complete picture of the future events. However, this approach requires further testing and evaluation through reliable winter gauging dataset that can be used in model calibration as well as performance evaluation.

Future work is focussed on increasing the event dataset to further test and optimise the system. It also focuses on the assessment of the steady state hydrodynamic simulation configured within the hydraulic modules of the system.

.



**Code and Data Availability**

The Delft-FEWS configuration of the system is available at https://zenodo.org/doi/10.5281/zenodo.11507599. This includes all the dataset configured into the system and used in this study. This version can only be used for
demonstration purpose. For research and operational applications license agreement must be signed with Deltares. The external models i.e. HOOPLA and HECRAS are also available in the configuration. Matlab Runtime (version 9.1) should be installed on the computer trying to run HOOPLA through this configuration. This is available as a free resource at https://www.mathworks.com/products/compiler/matlab-runtime.html. The observed meteorological and hydrometric data is the property of © Gouvernement du Québec, ministère de l'Environnement et de la Lutte contre
les changements climatiques, de la Faune et des Parcs, 2022. The authors of this paper declare responsibility for the observed data made public through this work.

**Author Contribution**

**Kh Rahat Usman:** Methodology, Framework development, Delft-FEWS configuration, analysis and writing. **Rodolfo Alvarado Montero:** Delft-FEWS configuration, writing. **Tadros Ghobrial:** Conceptualization,
Methodology, Supervision, Funding acquisition, Writing-review, and editing. **François Anctil:** Methodology, Supervision, Writing-review, and editing. **Arnejan van Loenen:** Delft-FEWS configuration, Writing-review and editing.

**Competing Interest**

The authors declare no conflict of interest.

**Acknowledgments**

This research project was funded by the Ministère de Sécurité publique (MSP) of the Québec Government under the FLUTEIS project (project number CPS 18-19-26). We would like to thank Pascal Marceau, and Jean-Philippe Baril-Boyer the MSP for their continuous support and collaboration. The authors would like to acknowledge the support by Dominic Roussel and Judith Fournier from the Ministère de l'Environnement, de la Lutte contre les changements
climatiques, de la Faune et des Parcs (MELCCFP) to provide relevant data as well as insights into their operational system to carry out this study.

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
