# Peer review of "Development of an under-ice river discharge forecasting system in Delft-Flood Early Warning System (Delft-FEWS) for the Chaudière River based on a coupled hydrologicalhydrodynamic modelling approach."

_Geoscientific Model Development, 2024_

## Author Comment (AC3)

**Major comments:**

I have several concerns about your model evaluation. In your forecasts for the evaluation, the measurement is always available on day 1. Why did you choose to do so? What is the difference if you start the forecast earlier (so that the measurement is available on day 2, 3, ...)? Does this help to evaluate the model forecast for different days? Could you perform this analysis?

The main consideration behind evaluating the model performance with measurement on the first day was to reduce the element of uncertainty arising from sources such as hydrological model states, initial conditions and observations. In the current study the aim was to test the performance of the coupled modelling framework with as minimum input data uncertainty as possible. Therefore, we decided to use meteorological observation data as forcing data instead of using meteorological forecast (hindcasts in this case) from the REPS. Similarly, to evaluate the hydrological model performance we deemed it beneficial to place the measurement on the first day as the initial conditions and model states uncertainty is lowered due to data assimilation and updated model states.

Furthermore, river ice conditions are dynamic and can change quickly over a short period of time. The river-ice hydraulic model is sensitive to two parameters: (i). Discharge; (ii). Ice thickness. Since the measured ice thicknesses are available on the day of measurement, it was a natural choice to start the forecast from the day of measurement and use the measured ice thicknesses for ice cover data in the hydraulic model.

The authors note and understand the concerns of the reviewer with regards to placing the measurement on the first day for each forecasting experiments and subsequent model performance evaluation. Therefore, we have conducted additional analyses following the recommendations of the reviewer. The additional analysis includes starting forecasting experiments 3-4 days before the day of discharge measurement at site and then evaluate the performance of the modelling setup using RMSE and PBias for different forecast lead times. This will be incorporated in the revised draft. However, this new analysis comes with an additional assumption of ice cover thickness remaining constant during the days leading up to the date of measurement.

Unfortunately, the discussion of your work is very limited. What are the implications and limitations of your work? How is your model performing better than models not accounting for ice conditions? Could you discuss the uncertainty in your data and model estimates as well as your system evaluation? How do you estimate the uncertainty of your corrected observed discharge data (corrected with the backwater coefficient)? How does your study compare to others, what are differences and similarities?

We acknowledge the point raised by the reviewer and understand that some important questions were not addressed in the discussion. We are committed to addressing these questions in the revised draft and improving the discussion.

You discussed one event in detail, including three figures, and provided both the results of the hydrological and hydraulic modelling. Why have you chosen this period? Is it representative of frequently occurring conditions?

The main objective of this paper is to demonstrate the details of the coupled system built to estimate under-ice discharge using coupled modeling. With this objective in mind, we dedicated a large portion of the paper for describing the system. Then, to showcase the performance of the system and how it operates, we chose a typical winter event to do so. Moreover, this event provided a unique opportunity to show case the value of coupled modelling for under ice discharge estimation. We addressed key points such as updating the hydrological states and the hydraulic model state before each new forecasts, and the effect of data assimilation in forecasting. When we compared the hydrologic forecasts with the corrected discharges, we observed significant deviations in this forecast. However, when we ran these forecasted hydrographs through the hydraulic model, we observed smaller deviations and good agreement with the shape of the observed water levels. This casts some doubts on the quality of the corrections done for estimating the under-ice discharge. Hence, we decided to present this single event to discuss the working of the system, as well as the uncertainty in the winter discharge estimates.

**Minor comments:**

How can the presented methodology be transferred to another study site? Which conditions and data are required? What may cause problems?

The current study is a first step towards developing a coupled (hydrological and hydraulic) modelling approach for continuous estimation of under ice discharge in cold climate regions where rivers undergo freeze-up during the winter months and conventional discharge measurement/estimation method become inapplicable. The current methodology can be transferred and applied to other study sites that experience extreme cold climate such as the one similar to Canada. However, it will require hydrological model(s) that best describe the target catchment's hydrology (must take into account solid precipitation accumulation & melt processes), and a well calibrated river ice hydraulic model. The data requirement will vary depending on the scope of the study, choice of modelling tools for the study and data availability in the study site however, basic requirements will be climatic data (precipitation and temperature), hydrometric data (water levels and discharge), river bathymetry and ice thickness data. The main challenge in this approach however remains with ice cover thickness data which needs to be collected from the field. We will address this issue in the Discussion.

This study strongly relies on Montero et al., 2023. Please clarify the difference between your study and the previous one as well as your advances compared to that study.

Montero et al. (2023) was a conference paper prepared for the 22nd Workshop on the Hydraulics of Ice Covered Rivers (2023) organised by Committee on River Ice Processes and the Environment (CRIPE). This conference paper *titled "Progress in the development of an operational testbed using Delft-FEWS for river ice forecasting on the Chaudière River, Quebec"* provided a preliminary report on the progress of the system which the authors had

started to develop. This work did not include evaluation of the system's performance or its application for under-ice discharge estimation or forecasting. The current work builds on this preliminary work and provides detailed insights on the components of the systems and evaluates the performance of the coupled modelling system for under-ice discharge estimation and forecasting based on field observations. Furthermore, data assimilation was discussed as a tool implemented in the system but was not used in hydrological modelling framework presented in Montero et al. (2023). The current study implements data assimilation in hydrological forecasts. We thus believe that the present work constitutes a significant progress over Montero et al. (2023).

L21: system or systems?

The word "system" is correct and will be corrected in the revised draft.

L29: Is the hydrological module better including the ice component than without it?

The hydrological module does not have an ice component as the river is not directly accounted for. The ice component in the natural system is modelled/simulated through the hydraulic module of the system. The hydrologic module consists of conceptual hydrological models producing runoff irrespective of river ice conditions. However, the hydrological modelling framework HOOPLA does have a snow accounting routine which takes into account the snow accumulation and melt processes in the catchment. The hydraulic module, consisting of a river-ice hydraulic model, helps compare the simulated water levels with the observed water levels as a check for the performance of the hydrologic module.

L36: introduce

The grammatical error (i.e. "introduces" in the current draft) will be corrected in the revised draft.

L50-58: How are measurements of under-ice discharge made in other countries across the world?

The other region influenced by river ice is the Nordics (Iceland, Norway, Sweden and Finland). A study (in French) conducted by Turcotte and Morse (2017) summarizes the practices for estimation of under-ice discharge in the Nordics. The overall procedure followed by the different agencies in Nordics responsible for maintaining hydrometric records is same. During the winters these agencies performs a single manual measurement for discharge at certain sites using conventional current meter. No real time data is produced during the winter. The winter flows are estimated through comparison hydrographs and hydrological forecasting. The experience of the operator is important for winter discharge data quality. The manuscript will be modified to include that information.

L61: What is the HBV model? Can you describe it quickly?

The Hydrologiska Byråns Vattenbalansavdelning (HBV) model was developed at the Swedish Meteorological and Hydrological Institute (SMHI) by Bergström (1995). It is a conceptual hydrological model that consists of routines describing snow accumulation and melt, soil moisture accounting, runoff and routing. The model runs on precipitation, temperature and

potential evapotranspiration. The HBV model runs as a semi-distributed hydrological model with basin divided into different zones based on altitude, lake area and vegetation (Zhang and Lindström, 1997). A more detailed description of the HBV model will be included to the revised draft.

L83: Please provide a reference for HEC-RAS.

The reference will be added to the revised draft.

Table 1: Please provide a reference for your data.

The reference will be added to in the revised draft.

Figure 1: Please clarify your figure (increase font size, no upside down labels, make clear that the Lower Chaudière is not included into the analysis, use different shapes and easily visible colors for the different measurement stations). You may consider labelling the subcatchments right next to them for simplified identification. Please add also a clearer map of the location of the study area in North America, including readable city names.

The figure will be revised according to the recommendations of the reviewer and guidelines of GMD.

L155: The correct name is Environment and Climate Change Canada.

The correction in the name will be made in the revised draft.

L156: Please provide the reference of these data sets, not of the secondary literature.

The reference will be added to the revised draft.

L157: Please provide the reference.

The reference will be added to the revised draft.

L170-186: Please clarify this paragraph, the determination of the backwater factor is not clearly described and easily understandable but has a major impact on the corrected data. How do you make sure that the applied correction is correct and representative of the in-situ conditions?

The backwater correction factor is estimated by MELCCFP through procedures developed internally. This involves an estimation of river ice conditions by experienced staff. The open water rating curve provides discharge estimates at a 15-minutes interval. This 15-minute discharge data is used to calculate an average discharge for the day. The backwater factor estimated by the experienced staff at MELCCFP is applied on this average discharge calculated from the rating curve for the day and is published as the corrected under ice discharge. We had obtained both the raw and corrected data from MELCCFP i.e. the corrected under-ice discharge at a daily timeframe and the raw (uncorrected) instantaneous discharge at 15-minute timeframe. We calculated the average daily discharge from the raw timeseries and then divided the corrected discharge with the open water daily average discharge to estimate the correction factor used by MELCCFP for each day of the season. Once we calculated the correction factor for each day, we applied it on the raw timeseries to

have a at the 15-minute interval. This assumes that within the day the ice conditions did not change significantly, and the correction factor is representative of the ice conditions throughout the day. The revised draft will include clarifications on the determination of the backwater factor.

In summary, the ministry of environment has internal protocol for estimating those coefficients based on local knowledge of rivers and river ice conditions, manual measurements and actual hydrometeorological conditions. Nevertheless, the objective of this paper is not to quantify uncertainty in public discharge data. This could be the focus of future research. The main objective of this paper is to demonstrate the feasibility of building a coupled model to estimate under-ice discharge.

Figure 2: Please increase the font size of the axis labels and label the subplots as a) and b) instead of using figure titles.

The figure will be revised according to the recommendations of the reviewer and guidelines of GMD.

L194-202: Please provide references.

The reference will be added to the revised draft.

L206-215: Please provide references for the used softwares and models. What is precisely the novel contribution of your study in this methodology and which part of this approach has been used in former studies?

We will add the following references to the paragraph: "The Delft-FEWS platform offers a flexible and configurable framework for storage and processing of hydro-meteorological data, as well as model integration and coupling hence, making it a simple platform choice for any forecasting system (Gijsbers et al., 2008; Werner et al., 2013)."

The novel contribution of our study relies on coupling the hydrological and the river ice hydraulic model to obtain an estimate of under-ice discharges. Moreover, this study enables the data assimilation of discharge data through the HOOPLA toolbox to improve model states at forecast time.

L218: Are only the resulting precipitation values used or also other data?

The text will be replace with: The Regional Ensemble Prediction System (REPS), operated by Environment and Climate Change Canada, provides a probabilistic prediction of precipitation and temperature (among many other atmospheric variables) over a 3-day forecast horizon. The REPS forecasts are generated by introducing small perturbations to the initial and boundary conditions of the model, which in return create slight variations in the prediction.

The REPS integrated into the current system provides both precipitation and temperature forecasts for hydrological modelling.

Figure 3: Is REPS0 the unperturbed control member? Why is it framed? Please write the numbers of the control members elsewhere in each subfigure instead of within the catchment. The color scheme of the legend seems to be different to the one used in the figures.

REPS0 was framed only because it was used in the software for further visualization options. We understand this could be confusing. We will replace the figure to remove the frame around it. Furthermore, we will move the numbers of the control members and correct the legend.

L240-242: Please clarify. Are all data (open water and ice) used at a 3h resolution for the modelling? If yes, how have you increase the resolution during ice conditions, where only daily data is available?

During model calibration and validation, the discharge data fed to the model was in 3h resolution for the entire period considered i.e. from 2008 to 2012 (calibration) and 2014 to 2018 (validation). Since during the winter months discharge data is not available in a finer temporal resolution (for instance every hour or every 3 hours) but is rather available at daily resolution (i.e. one discharge value reported for the entire day), we decimated the daily discharge data to 3-hour discharge using linear interpolation. The revised draft will provide clarifications.

Figure 4: Please increase the font size of the axis labels and label the subplots as a) and b) instead of using figure titles. The data are hardly visible, so that the measurements and the model are hardly distinguishable. This hinders the evaluation of the model's performance. Please improve the readability of the results.

The figure will be revised according to the recommendations of the reviewer and guidelines of GMD.

Figure 5: Please increase the font size of the axis labels.

The figure will be revised according to the recommendations of the reviewer and guidelines of GMD.

L262-265: What do you think is the reason for the lower KGEm in the validation case in the Upper Chaudière?

The modified Kling-Gupta Efficiency (KGEm) (Kling et al., 2012) consists of three statistical properties: (i) correlation; (ii) bias ratio; and (iii) variability ratio. Analysis of the KGEm for Upper Chaudière showed that during the validation period the calibrated models had bias ratios greater than 1, as well as higher than those during calibration period indicating overestimation of discharge during the validation period. The variability ratio was lower than 1, as well as lower than the values obtained during the calibration period, indicating that the ability of the models to capture variability during the validation period was slightly weaker than during calibration. This led to reduced KGEm value during the validation period. The correlation coefficient for both calibration and validation period was comparable showing that the models were reproducing the shape of the hydrograph efficiently, however the

magnitude of simulated discharges were slightly off as indicated by bias ratio. This point will be included in the revised draft.

We also analyzed the meteorological conditions using average monthly precipitation and temperatures (Figure 1) during the two periods i.e. calibration and validation and found that there was no significant difference in the amount of average annual precipitation. However, there was a slight redistribution of precipitation during the validation period with more solid precipitation as compared to calibration period which could have contributed to overestimation of flows during the winter months. Moreover, average monthly temperatures were also relatively lower during the validation period as compared to the calibration period this could have also contributed to reduced evapotranspiration calculations in the HOOPLA framework leading to flow overestimations and hence relatively lower model performance.

[Figure]

*Figure 1 Meteorological Analysis of Upper Chaudière for the calibration and validation periods. The data represents monthly averages calculated over the 5-years during each period (i.e. Calibration and Validation) from meteorological timeseries used.*

L267-270: Please explain why you are using the unsteady model.

We can only feed timeseries hydrographs to unsteady models. Using an unsteady river ice hydraulic model is in lines with the aim of the study i.e., is to develop a mechanism for continuous estimation of the under-ice river discharge during the winter season. This includes periods leading to the formation and release of ice jams where a steady state model is inappropriate for such applications (Hicks and Healy, 2003). Moreover, an unsteady river ice model provides useful insights on water level fluctuations due to flow variability, renders the application of flow hydrograph easier and simulates the river-ice interaction effectively.

L483-484: To which water stage does this elevation correspond? How many days per year or measurements are not possible at this site? What is its influence? Can measurements be performed in winter during the ice conditions?

We assume you refer to lines 481-482. The location under consideration is Valee Junction which is a Comité de bassin de la rivière Chaudière (COBARIC) operated water level monitoring station. This station does not have an open water rating curve developed for discharge measurement therefore, it is difficult to comment on what exact value of open water discharge corresponds to the threshold elevation of 142.96 m. The probe for measuring the water level is placed in a well and is unable to measure water level when the water is below the probe. The data that we currently have for this station spans from February 2018 to July 2023. The data can present up to 2 months of flat line corresponding to 142.96 m from January to March in some years (for example in the year 2021 the data presents a flat line for nearly 60 days). The impact of this is that it is difficult to trust this station as control point in the hydraulic model and therefore, it was removed from evaluation of the model. COBARIC continuously uses this station for monitoring Chaudière even under the river ice conditions, since their mandate is flood protection in the area, the threshold of 142.96 m is not an issue for them and most likely this threshold corresponds to a smaller discharge value.

L494: How are the RMSE and Pbias calculated for the average ensemble values for each event? Are they calculated for every ensemble and then averaged to obtain the results in Figure 13?

For RMSE the squared error for each ensemble member at each time step was calculated and then the mean was calculated to get the RMSE of the ensemble at each time step. Similarly, Pbias for each ensemble member was calculated and then averaged to calculate the average Pbias of the ensemble. After obtaining RMSE and Pbias for each event, an average over all four simulated events was calculated to get an average RMSE and Pbias for the system which was presented in Figure 13. The draft will be revised to provide more details.

L500, 508: Please reorder figures or their reference, here Figs. 11 and 12 are referenced before Fig. 10.

This will be addressed in the revised draft.

L502-504: How was the discharge measured? How is the averaged discharge for the entire day determined?

MELCCFP performed discharge measurements. MELCCFP uses the mid-section method for discharge measurements. The mid-section method for measuring under-ice discharge involves dividing the cross-section into panels. Holes are drilled through the ice cover for each panel and a velocity profiler (such as an ADV or ADCP) is lowered through these holes to obtain an average velocity for the panel. Discharge through each panel is calculate by multiply velocity with the under-ice panel area. The sum of all the panels gives discharge through the entire cross section. For average daily discharge the instantaneous discharges obtained from the rating curve are averaged over the entire day to obtain a daily discharge. This is then corrected using a backwater correction factor estimated by the expert operators

who have good understanding of local river ice conditions and the backwater effect and are aided by the manual discharge measurements.

L524: You mentioned that ice thickness and the under ice roughness are important. Have you performed any measurements of the under ice roughness? If no measurements are available, how is this value chosen in the model?

We have performed some simulations to study the sensitivity of ice roughness on the simulated water levels. We found that water level simulations are more sensitive to ice thickness and discharge when compared to ice roughness. In this study we followed the ice roughness values recommended in HEC-RAS hydraulic reference manual (Brunner, 2016) based on the assumption that the ice cover is rippled ice sheet.

L504: How was the ice thickness measured? At one cross-section only or several ones?

For the February 2020 event we had detailed ice measurements along the entire reach available from the work by Ghobrial et al. (2023) and these were used for the model. For other events, measured ice thicknesses from MELCCFP taken at the cross section where the discharge was measured, were used for the entire river. This could induce some additional uncertainty at Sainte Marie station.

Figure 10: Please label the subfigures with according the journal guidelines, instead of using figure titles. Please provide the reference for the used data.

The figure will be revised according to the recommendations of the reviewer and guidelines of GMD. The data used for producing the figure is from the sources mentioned in the section 2.2 Data of the paper.

Figure 11: Please clarify this figure. Especially in the right part (after Feb 25[th]), the different lines and percentiles are difficult to distinguish, you may want to use different line styles and filling styles.

The figure will be revised according to the recommendations of the reviewer and guidelines of GMD.

L531-545: Please consider splitting this paragraph. What is the uncertainty in the corrected observed data?

The paragraph will be reworked in the revised version. The uncertainty in the corrected data is not fully known. Laval University team has conducted some manual discharge measurements along the Chaudière and Famine rivers sites. The comparison of these measurements with the published discharge (i.e. the discharge corrected by MELCCFP and published on their website) showed differences in the range of -4% to -37% of the measured discharge for Famine, where the negative sign indicates that the published discharge was lower than the measured discharge. While for Upper Chaudière the differences ranged from -34% to +27% of the measured discharge. These are based on 6 manual measurements on each site during February to March 2023.

Figure 12: Please also rework this figure.

The figure will be revised according to the recommendations of the reviewer and guidelines of GMD.

Figure 13: Please use different colors in the top and bottom plots to represent different locations. What represents the grey shaded areas precisely? Is this performance evaluation for this specific February 2022 event only or averaged values for all events? Please clarify.

The figure will be revised according to the recommendations of the reviewer and guidelines of GMD. The performance evaluation is averaged over the 4 events. The grey shaded areas represented the 90% confidence intervals determined through bootstrap method.

L654-655: Are those your future steps or do you generally recommend this to others?

These are the future steps for the current team developing the framework. We aim to build on this forecasting system and methodology. The current work is a foundational work for the proof of concept of applying coupled hydrological and hydraulic modelling for continuous estimation of winter discharge. We aim to further develop this approach and the modelling setup so that it can be applied within operational forecasting environments.

**References:**

Bergström, S.: The HBV model, Computer models of watershed hydrology., 443-476, 1995.

Brunner, G. W.: HEC-RAS River Analysis System Hydraulic Reference Manual, U.S. Army Corps of Engineers Institute for Water Resources, 2016.

Ghobrial, T., Pelchat, G., and Morse, B.: A comprehensive field investigation of the dynamic break-up processes on the Chaudière River, Quebec, Canada, Hydrology Research, 54, 797-817, 10.2166/nh.2023.137, 2023.

Gijsbers, P., Werner, M. G. F., and Schellekens, J.: Delft FEWS: A proven infrastructure to bring data, sensors and models together Proceedings of the iEMSs Fourth Biennial Meeting: International Congress on Environmental Modelling and Software, Barcelona, Catalonia, 28-36,

Hicks, F. E. and Healy, D.: Determining winter discharge based on hydraulic modeling, Canadian Journal of Civil Engineering, 30, 101-112, https://doi.org/10.1139/l02-031, 2003.

Kling, H., Fuchs, M., and Paulin, M.: Runoff conditions in the upper Danube basin under an ensemble of climate change scenarios, Journal of Hydrology, 424-425, 264-277, https://doi.org/10.1016/j.jhydrol.2012.01.011, 2012.

Montero, R. A., Usman, K. R., Ghobrial, T., and Van Loenen, A.: Progress in the development of an operational testbed using Delft-FEWS for river ice forecasting on the Chaudière River, Quebec, CGU HS Committee on River Ice Processes and the Environment 22nd Workshop on the Hydraulics of Ice-Covered Rivers, Canmore, AB 2023.

Turcotte, B. and Morse, B.: Identification de méthodes visant l'amélioration de l'estimation du débit hivernal des cours d'eau du Québec, MELCCFP, 2017.

Werner, M., Schellekens, J., Gijsbers, P. J. A., Dijk, M., Akker, O., and Heynert, K.: The Delft-FEWS flow forecasting system, Environmental Modelling & Software, 40, 65-77, https://doi.org/10.1016/j.envsoft.2012.07.010, 2013.

Zhang, X. and Lindström, G.: Development of an automatic calibration scheme for the HBV hydrological model, Hydrological Processes, 11, 1671-1682, https://doi.org/10.1002/(SICI)1099-1085(19971015)11:12<1671::AID-HYP497>3.0.CO;2-G, 1997.